# Virosaurus A Reference to Explore and Capture Virus Genetic Diversity

**DOI:** 10.3390/v12111248

**Published:** 2020-11-01

**Authors:** Anne Gleizes, Florian Laubscher, Nicolas Guex, Christian Iseli, Thomas Junier, Samuel Cordey, Jacques Fellay, Ioannis Xenarios, Laurent Kaiser, Philippe Le Mercier

**Affiliations:** 1Vital-IT Group, SIB Swiss Institute of Bioinformatics, 1015 Lausanne, Switzerland; Anne.Gleizes@sib.swiss (A.G.); thomas.junier@epfl.ch (T.J.); 2Division of Infectious Diseases, Geneva University Hospitals, 1205 Geneva, Switzerland; Florian.Laubscher@hcuge.ch (F.L.); Samuel.Cordey@hcuge.ch (S.C.); Laurent.kaiser@hcuge.ch (L.K.); 3Laboratory of Virology, Division of Infectious Diseases and Division of Laboratory Medicine, University Hospitals of Geneva & Faculty of Medicine, University of Geneva, 1205 Geneva, Switzerland; 4Bioinformatics Competence Center, University of Lausanne, 1015 Lausanne, Switzerland; nicolas.guex@unil.ch (N.G.); christian.iseli@unil.ch (C.I.); 5Laboratory of Microbiology, University of Neuchâtel, 2000 Neuchâtel, Switzerland; 6Unité de Médecine de Précision, CHUV, 1015 Lausanne, Switzerland; jacques.fellay@epfl.ch; 7School of Life Sciences, EPFL, 1015 Lausanne, Switzerland; 8Host-Pathogen Genomics Laboratory, Swiss Institute of Bioinformatics, 1015 Lausanne, Switzerland; 9Center for Integrative Genomics, Faculty of Biology and Medicine, University of Lausanne, 1015 Lausanne, Switzerland; ioannis.xenarios@unil.ch; 10Geneva Centre for Emerging Viral Diseases, Geneva University Hospitals, 1205 Geneva, Switzerland; 11Swiss-Prot Group, SIB Swiss Institute of Bioinformatics, 1011 Geneva, Switzerland

**Keywords:** database, complete genome, bioinformatics, HTS, diagnostics, sequencing, viral infections, viruses

## Abstract

The huge genetic diversity of circulating viruses is a challenge for diagnostic assays for emerging or rare viral diseases. High-throughput technology offers a new opportunity to explore the global virome of patients without preconception about the culpable pathogens. It requires a solid reference dataset to be accurate. Virosaurus has been designed to offer a non-biased, automatized and annotated database for clinical metagenomics studies and diagnosis. Raw viral sequences have been extracted from GenBank, and cleaned up to remove potentially erroneous sequences. Complete sequences have been identified for all genera infecting vertebrates, plants and other eukaryotes (insect, fungus, etc.). To facilitate the analysis of clinically relevant viruses, we have annotated all sequences with official and common virus names, acronym, genotypes, and genomic features (linear, circular, DNA, RNA, etc.). Sequences have been clustered to remove redundancy at 90% or 98% identity. The analysis of clustering results reveals the state of the virus genetic landscape knowledge. Because herpes and poxviruses were under-represented in complete genomes considering their potential diversity in nature, we used genes instead of complete genomes for those in Virosaurus.

## 1. Introduction

At least 130 different viruses infecting humans have been recorded so far [1]. Diagnostic assays rely on specific procedures for each virus screened, mostly by using enzyme-linked immunosorbent assay (ELISA) or molecular techniques such as real-time polymerase chain reaction (qPCR) [2]. These tests are robust but are, by design, restricted to what the researcher is looking for. Moreover, some viruses resist diagnostics by these methods because of their unusual genetic diversity, such as Lassa virus [3]. Classical diagnosis methods are adapted in routine diagnosis, but show limitations for some research, such as viral surveillance or virus discovery investigations that require testing all known viruses in one sample [4,5].

High-Throughput Sequencing (HTS) technology makes possible the detection of all genetic entities present in a sample. Thus, this method offers promising perspectives for virus detection and diagnosis [6]. HTS produces millions of reads that are identified by comparison with a virus reference dataset. Many laboratories have set up their HTS virus diagnostic pipeline by building up their own reference database, mostly by cherry-picking virus reference sequences from NCBI/GenBank or UniProt resources [7,8,9,10]. Building a consistent reference database is a challenge because viruses are extremely diverse and variable [11,12]. Creating a non-biased virus reference database requires comparing all viral genomic material to identify the minimum number of sequences needed to cover all genetic diversity at a given percentage of identity [13].

To fill these requirements, Virosaurus (for Virus Thesaurus) is an unbiased viral database built from eukaryotic virus genomes publicly available, cleaned and annotated. The focus of this database is not only to provide accurate references of virus diversity, but also to add a layer of annotation for better clinical analysis of results. Other virus reference databases have been published but with a different scope or annotations. VirusSITE is a virus reference database published in 2016 [14], mainly based on NCBI RefSeq sequences [15]. In 2018, Arifa Khan’s group at the Food and Drug Administration (FDA) released the Reference Viral Database (RVDB) for virus detection using high-throughput sequencing [16], which is also based on RefSeq [15], and enlarged by a similarity search. Virosaurus has a different approach, being directly built on all GenBank sequences with the addition of a layer of clinical annotation.

Here, we describe the Virosaurus reference database and the method used to build and annotate it. We focused on complete virus genomes to be compatible with existing clinical pipelines [7,8,9,10], and to allow better quality control in the output. All sequences are clustered to remove redundancy and reduce the size of the database, thereby improving its effectiveness for computational analysis without reducing its quality. An annotation layer is added to all sequences in order to facilitate direct analysis in the context of clinical metagenomics. A “virus” is a biological entity whereas a “species” or “genus” is a taxonomic classification. There is no available controlled vocabulary for ‘virus’ names associated with sequence data; therefore, we used mainly “species” and “genus” to identify viral sequences. Throughout the manuscript, we use “virus species” in a looser sense, and refer to “ICTV species” wherever applicable. Sequence names are normalized at the level of virus genera, species, or genotypes in accordance to the clinical relevance of each entity. This allows the grouping of all HTS reads under a virus species in order to have a quick overview of pathogens present in the sample. When sequences had to be classified under species level, we referred to it as “sub-type”. Furthermore, each virus genome is annotated as DNA, RNA, circular or linear to enable different analyses of HTS reads as needed. Curated annotation adds important value to Virosaurus by providing critical and accurate information for results interpretation. The pipeline used to construct Virosaurus is automatized to facilitate regular updates.

## 2. Materials and Methods

All virus data (taxonomy, sequences, clusters and annotations) were stored in a relational database (PostgreSQL) with a Java written interface. FASTA files were extracted from this database.

### 2.1. Data Collection and Cleaning Up of Viral Sequences

All viral sequences were downloaded from the INSDC database at GenBank in FASTA file format. Because we aimed at creating a eukaryotic virus database, these sequences were filtered using the NCBI taxonomy hierarchy. Virus sequences belonging to genera able to infect eukaryotes as per ViralZone (https://viralzone.expasy.org/655) have been kept and divided in vertebrate, plant and other eukaryotes, the others have been discarded. The list of vertebrate host viruses comprised 330 virus genera. Viral species were assigned to each sequence using the hierarchy of NCBI taxonomy, and the official ICTV species names (https://talk.ictvonline.org/taxonomy/). Low quality sequences were then removed through a series of quality controls: removal of sequences with too many “N”s (more than 10%) or at least one gap annotation of unknown length. Some sequences classified under viral taxonomy contained non-viral elements, like vectors or recombined oncogenes. Since those are liable to produce false positives during the analysis, they were removed by the application of keywords filtering (see main text). All quality controls were refined after testing against at least 10 known clinical blood samples at each release to identify false positives.

### 2.2. Generation of Complete Virus Sequence Dataset

The first step required identifying non-segmented complete virus genomes, or complete segments for segmented viruses. “Complete” is defined here as a sequence containing at least all coding regions. To do so, all the data downloaded from GenBank were first prepared by removing all obviously incomplete sequences through keywords like “partial”, “incomplete”, “near complete”, etc. To assess completeness of sequences, a range of sizes was manually curated for each virus using all available data in INSDC [17]. The sequences were separated into two pools: non-segmented genomes and segmented genomes. Non-segmented genomes in which the size fell within the range for complete genomes were labeled as such. For segmented viruses, the difficulty came from the fact that segments are sometimes not annotated or miss-annotated in the INSDC database. The segment names that we used are those reported in ViralZone, and come from ICTV approved proposal documents. All segmented virus sequences were clustered and the segment name was identified by checking for similarity against UniProt (BLAST). All segments have been processed exactly as monopartite genomes for quality and completeness. Once all segment names were identified by sequence similarity, their completeness was estimated by comparing to a matrix of complete segment size ranges for each genus. Size data were established manually for each virus genus using INSDC data. The complete non-segmented genomes and complete segments were pooled to create our complete virus sequence dataset.

### 2.3. Clustering

The Virosaurus database was clustered using CD-HIT [18]. Complete virus sequence data were clustered in 29 h (90% identity) and 1 h 30 min (98% identity). In our first test, we also clustered herpesvirid and poxvirid complete sequences, but those were quite long to compute because of the large size of these viruses (46 h for 90% identity). Replacing those big sequences by virus genes sped up the clustering computation (9h at 90% identity, 3 min at 98%). Clusters were then controlled for size homogeneity (cut-off 80%) and checked if they referred to a single species. The few clusters comprising sequences from more than one species were checked manually. Some multi-species clusters were due to user deposition mistakes, for example Torque teno sus virus 1a instead of 1b, and this was corrected. Others are sequences similar to other members of the cluster but classified under different species. For those, all names were reported in the FASTA header representing the cluster.

### 2.4. Generation of Virus Genes Dataset

All herpes and poxvirus genes were identified from cleaned up GenBank sequences based on GENE feature annotation.

### 2.5. Clinical Annotation

Genotype or serotype data were identified by using GenBank taxonomy classifying *Norwalk*, *Dengue*, and *Hepatitis C* viruses. Sub-types classifications relevant to medical diagnostics were added by using NCBI taxIds identifying Enterovirus 71, Enterovirus 68, low-risk or high risk HPVs, Polio or non-polio enterovirus C, and novel or classical for mamastrovirus. These data were labeled “clinical typing” in the FASTA file. Another field called “usual name” was created to include common naming for viruses having several names (Merkel cell polyomavirus = Human polyomavirus 5). Papillomaviruses and torquetenoviruses were not considered relevant in a clinical report, so for those the usual name is “HPV” or “TTV” in order to pool all these viruses into one entity.

### 2.6. FASTA Format

Annotations are stored in FASTA header. The header contains 11 different topics annotated by a controlled vocabulary. Data come from GenBank, ICTV, ViralZone and manual curation.

<GenbankID> Genbank accession number of the sequence displayed in the FASTA.

<SequenceID> By default Genbank accession number of the sequence displayed in the FASTA. If the displayed sequence is a portion of a GenBank entry, for example in the case of genes, the Sequence ID is a unique identifier like GENE_583-3988.

Usual name = Name of clinical level entity; if the scientific name is not commonly used, the common clinical name replaces it, for example parvovirus B19 is the usual name of the Primate erythroparvovirus 1 species. Clinical level = Gives the taxonomic level suggested to be relevant for usual clinical diagnostics. By default <species>, but can be at <genus> level like for TTVs or HPVs.

Clinical typing = Unknown by default. Otherwise contains data clinically relevant below the species level. This can be genotypes (example:HCV) or qualifiers (polio enterovirus, High risk HPV, etc.). In rare cases of mixed clusters, several types are listed separated by a comma, this notably happens for some HPVs “low risk” and “undetermined risk” that are very similar.

Species = indicates the current official species name, as reported by the International Committee on Taxonomy of viruses (ICTV): https://talk.ictvonline.org/taxonomy/. In rare cases of mixed clusters, several species are listed separated by a comma, this notably happens for some segments of rotavirus A and C, which are very similar within different species.

Taxid = Taxonomy identifier from NCBI taxonomy database: https://www.ncbi.nlm.nih.gov/taxonomy of the taxonomic entity at species level.

Acronym = Abbreviation referring to virus name as reported in ViralZone acronym list: https://viralzone.expasy.org/resources/Acronyms.xlsx Nucleic acid= Nature of viral genome, either RNA or DNA for most viruses, RNA/DNA for retro-transcribing viruses (Ortevirales).

Circular = Y or N for yes or no. This is essential to efficiently map reads at both extremities of the FASTA sequence.

Segment = N/A for non-segmented viruses. For segmented genomes: official segment name as reported in the ViralZone database: https://viralzone.expasy.org/

### 2.7. Evaluation on Clinical Samples

During development of the database, beta versions have been tested using an unpublished pipeline (https://github.com/sib-swiss/virusscan) on a set of sequencing data (Illumina HiSeq 2500/4000 instruments (Illumina, San Diego, CA, USA)) obtained from 20 human clinical specimens (including plasma, serum, bronchoalveolar lavage, nasopharyngeal swab and cerebrospinal fluid specimens). These samples were previously reported positive by HTS for RNA and/or DNA viruses known to infect humans [19,20] using an updated version of the ezVIR pipeline [9]. The size of Virosaurus is about 1.6 gigaoctets (Go) unclustered, but is down to 243 or 107 megaoctets (Mo) for Virosaurus90 and 98 respectively. The small size of clustered data allows quick computation on samples.

## 3. Results

### 3.1. Data Collection

Virosaurus has been created using all GenBank genetic data (release 19-03-2020) (Figure 1). This method is unbiased, as it does not use pre-existing reference sets. Sequences were collected from all viruses belonging to families infecting either vertebrates, plants or other eukaryotes (insects, fungus, etc.) (https://viralzone.expasy.org/655). Non-vertebrate eukaryotic viruses might be useful for research or non-clinical studies. We have identified 330 virus genera in which at least one virus infects vertebrates (https://viralzone.expasy.org/655). All sequences have been quality-controlled to exclude low-quality data with gaps or stretches of “N”. Particular care has been devoted to the elimination of non-viral sequences that happen to be classified under “virus” in GenBank: sequencing or assembly artifacts, viral expression vectors, patented sequences, artificial recombinant sequences, and so on. These non-viral sequences can generate false positives, for example by being similar to human genomic fragments present in most clinical samples. Such erroneous sequences are removed by two means: sequences entries containing “patent”, “Vector”, “Artif*”, “recombinant” were excluded; and testing of the database on different types of characterized clinical samples allowed removing sequences producing false positives. In the 2020_04 release, 3,002,470 virus sequences have been recovered from GenBank, and 2,936,380 sequences have been selected by the pipeline for sequence completeness.

Identifying complete genomes was more challenging for segmented viruses, composed of two to twelve genetic segments. The complexity lies in the fact that segment naming is not consistent in GenBank deposition, and isolate names happen to be different across all segments. For those viruses, we have decided to work with complete segments instead of complete genomes. To harmonize and correct segment naming, all segments have been identified by BLAST against Swiss-Prot, and named using official nomenclature. Then, minimum and maximum size range of complete segments has been manually defined as above. Although heuristic, this method gives optimal results with the current data. Moreover, it is automated, which is an asset for updating the database.

Putting together complete genomes for non-segmented viruses, and complete segments for segmented viruses, creates the dataset we call the “complete virus sequence dataset”.

### 3.2. Clinical Annotation

The complete virus sequence dataset has been annotated by adding controlled vocabulary metadata in the FASTA files (Figure 2). All sequence taxonomy has been normalized to the species level, thereby matching reads can be classified under one viral entity (Figure 3).

Metadata about the viral genome comprises GenBank accession number, nucleic acid nature, number of segments, circular/linear and sequence. Considering that some sample preparation protocols target either RNA or DNA, it is essential to know if the viral genome is DNA or RNA for virus detection. For example, RNA virus detection in DNA preparations should trigger further investigations to rule-out contaminants. Whether the genome is circular or linear matters if users want to assemble HTS reads. In the case of circular genomes, reads can overlap at the start and end of the sequence. All metadata are added to the FASTA files in the header section of each sequence (Figure 2).

Each sequence in the INSDC database is linked to a virus isolate. Nonetheless, isolate data are not useful for clinicians in daily routine. To propose a relevant output, the Virosaurus database was designed to offer a broader classification of viruses at a higher level (Figure 3). We have used taxonomic classification: all sequences have been classified at the species level and annotation has been added about clinical virus belonging to this taxa: species name, TaxonomyID, acronym, usual name, clinical level and typing. The “usual name” displays the common virus name, which can be easier to interpret and more stable than the official name. For example, parvovirus B19 is the common name of a virus belonging to Primate erythroparvovirus 1 [21], most people are more familiar with the first than the latter name. Moreover, some viruses classification is not directly relevant to a clinical interest: there are 29 torque teno virus named Torque teno virus 1 to 29 and several of them can infect the same person [22]. Having a list of TTV<numbers> was not relevant for clinicians: the entire Torque teno virus received the common name “TTV”. The same has been done for alphapapillomaviruses, which have all “HPV” as “usual name”. These exceptions are signified in the field “clinical level”, describing the taxonomy level for which these names have been assigned to sequences. The annotated field “clinical typing” comprises genotype (ex: HCV; Norwalk virus), serotype (ex: Dengue virus) or disease (ex: polio, High risk HPV) when these data are necessary for some viruses.

### 3.3. Clustering

Because virus genomes are small and thus relatively easy to sequence, thousands of genome sequences are available, and these datasets are highly redundant. Redundancy can result in bias, and increases computational time with no benefits. Clustering computer programs are designed to removing redundancy by alignment methods. This consists of calculating clusters of similar sequences at a given percentage of similarity, and selecting one sequence to represent the cluster. We used CD-HIT to cluster our complete virus sequence dataset at 90% and 98% similarity thereby creating Virosaurus90 and Virosaurus98 [18]. Clusters were then reviewed to validate the quality of data. Ideally, each cluster should contain only sequences belonging to a single species. Clusters comprising sequences from more than one species were rare and manually checked. This low intermixing underlines the quality of taxonomic classification by ICTV [23].

Figure 4 displays data about the 13 most sequenced human viruses in GenBank. Interestingly, the relative composition of the datasets changes drastically between complete virus sequences, Virosaurus98 and 90 (Figure 4C,D). Influenza virus sequences are reduced 25 times by clustering at 98%, revealing a high redundancy, whereas HIV-1 sequences are reduced 1.3 times. Clustering at 90% reduces the number of influenza sequences 760 times, whereas HIV-1 sequences are reduced 2.8 times. This indicates that we have highly redundant sequences of influenza viruses in which the diversity drops at 90% similarity. In contrast, HIV-1 complete genomes are less redundant, but with a higher diversity. HIV recombinants might play a role in this observed diversity [24] as CD-HIT aligns full genomes and will create new clusters for most recombinants.

### 3.4. Reducing False Negatives in Virus Detection

Detection of viruses by HTS and bioinformatics analysis depends on the quality of the references. If the latter is suboptimal, this could lead to false negatives in the detection results for important viruses. Do we have enough knowledge of each viruses’ genetics to detect circulating viruses? The clustering method allows estimation of the genetic variability of a virus. A virus for which the genetic landscape has been thoroughly mapped will comprise many sequences in each cluster. On the other hand, we have only partial knowledge when clusters are isolated and contain few or a single sequence. Figure 5 shows the percent reduction of sequences by clustering at 90%. A low percentage suggests that we do not have significant data about the genetic landscape of a given virus. Some viruses like Influenza A virus, Dengue virus, Zika virus or ebolaviruses display a reduction of over 98%, meaning that we have at our disposal many more sequences than we probably need to detect those viruses in HTS samples. Of course, new strains may emerge but at least we have sampled much of the prior genetic data. Herpesviruses and poxviruses are the only viruses displaying a reduction inferior to 50%. In other words, the majority of clusters contain a single sequence. The worst cases are Human betaherpesvirus 6A and 6B for which clustering does not reduce the number of sequences, demonstrating a complete absence of redundancy in the current databases.

This lack of redundancy suggests that herpesviruses and poxviruses are more diverse than we know, and we might be unable to detect all circulating viruses with our reference dataset. We decided to use a different method to gather sequences for these two virus families. Since there are too few complete genomes, we have used genes instead of genomes (Figure 1). Using genes instead of genomes allows accessing many more sequences and maximizes the chances of identifying a virus with the reference dataset. Because the objective of Virosaurus is to analyze genetic entities at the level of species, using individual genes does not change the output post-processing.

### 3.5. Validation on Clinical Samples

The Virosaurus database 2018_11 has been tested on sequencing data from 20 human clinical samples previously characterized by HTS using the ezVIR pipeline (Figure 6) [9,19,20,21,22,23,24,25]. The results of this comparative analysis show 100% concordance. Moreover, the database has been used in a pilot study to characterize viruses causing fever, and provides a surveillance tool for emerging viral diseases in Gabon [26]. The results are concordant with the PCR screening results from a previous study [27].

### 3.6. Virosaurus Availability and Updates

Virosaurus can be downloaded online at https://viralzone.expasy.org/8676 and is licensed Attribution-NonCommercial-NoDerivatives 4.0 International (CC BY-NC-ND 4.0). We plan to update the database every year with new sequences, and to apply all changes in the taxonomy. The pipeline to create the database has been automatized, thereby lowering the cost and time of such an exercise. Moreover, we plan to apply quick hotfixes whenever erroneous sequences are reported, or when a new virus emerges. For example, the SARS-Cov-2 isolate Wuhan-Hu-1 complete genome was first added by patch in January 2020, and then fully included in the April 2020 release.

## 4. Discussion

The creation of a eukaryotic virus reference database using taxonomic clusters requires a list of complete virus genomes and segments. There was no database of complete virus genomes available; therefore, we developed a method to identify those genomes. This dataset of complete virus sequences is the first to be created, to our knowledge, and may be useful for further references or epidemiologic studies. By using complete genomes or segments, Virosaurus allows seamless integration with many existing pipelines in hospitals and helps the final interpretation of results. The underrepresentation of herpes and poxviruses among complete sequences led to the use of genes instead of genomes to broaden the genetic diversity covered by the reference dataset without affecting the methods used and the final output. Indeed, all reads are mapped to reference sequences, which are in turn mapped to a virus species or genus. Whether the sequences are complete genomes or single genes does not matter. The only inconvenience is that, for these two virus families, we lost the possibility of tracking the genome coverage by HTS reads. This metric is invaluable to assess artifacts or contamination, but is not mandatory. Indeed, PCR methods rely on the specific detection of only a small part of viral genomes and are accurate.

It is essential to minimize the risk of false negatives and false positives in diagnosis. False negatives could partially result from missing essential information. We estimated the quantity of references available for most human viruses and it revealed that herpes and poxviruses did not reach an adequate threshold. This highlights the need to sequence additional genomes for a better coverage of these viruses’ genetic landscape. Many false positives were easy to spot because of the similarity with the human genome and therefore were detected in most samples. We observed those false positives resulted mostly from sequences containing non-viral genes that were not filtered out by our method. False positives need to be evaluated at each release and manually excluded from the database. Another way to deal with sequences similar to human sequences would be to cross-reference the human genome with the virus database and exclude sequences from the latter, but this may remove too many sequences because of sequences of viral origin within the human genome.

The dataset of virus infecting vertebrates should be valuable for clinical diagnosis. In the last 100 years all emerging viruses have jumped to humans from vertebrate host reservoirs (e.g., Influenza, HIV, SARS, MERS, Ebola, SARS-CoV-2). Nonetheless, it is not impossible that some viruses belonging to a genus only known to infect non-vertebrate hosts do indeed infect humans in unknown conditions [28]. Moreover, virus discovery by metagenomic analysis does not need to be restricted to viruses infecting vertebrates. We have started to develop Virosaurus data for non-vertebrate viruses to cover those topics.

## Figures and Tables

**Figure 1 viruses-12-01248-f001:**
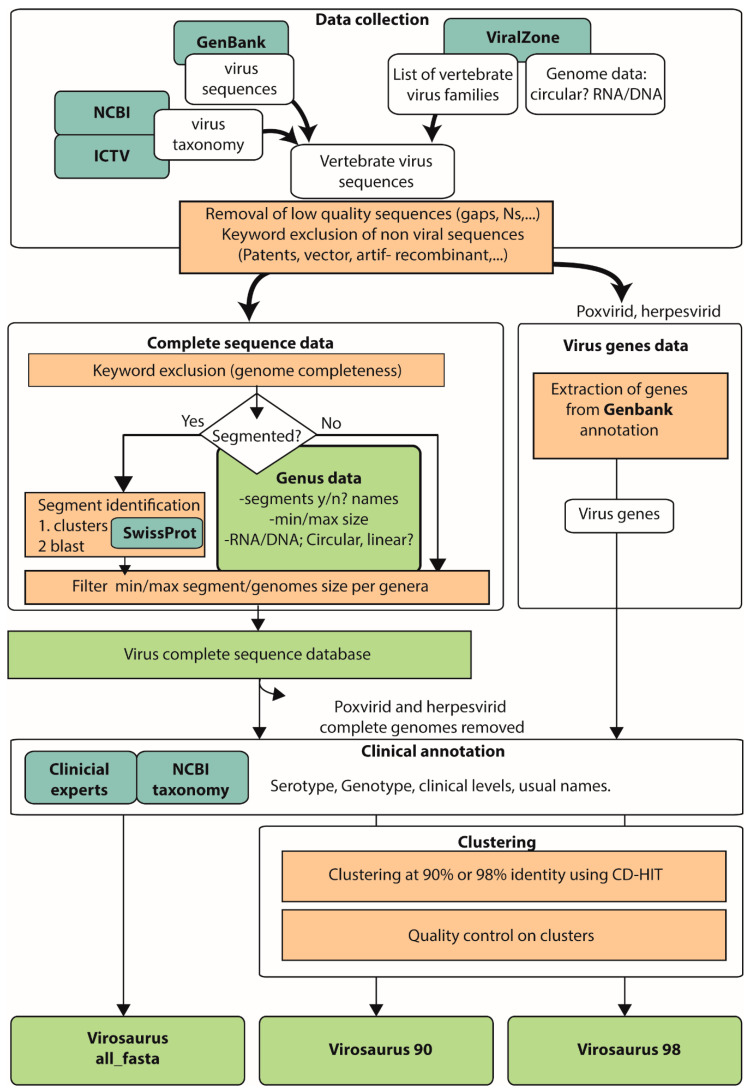
Workflow for the creation of the Virosaurus datasets. References are in blue, and output datasets are in green.

**Figure 2 viruses-12-01248-f002:**
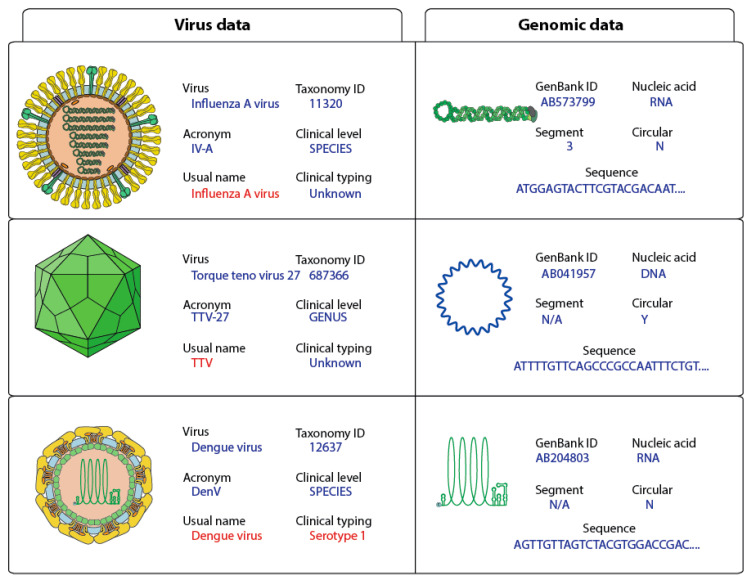
Examples of virus genome annotation. The usual name and clinical typing should be the default output for clinical studies and are shown in red.

**Figure 3 viruses-12-01248-f003:**
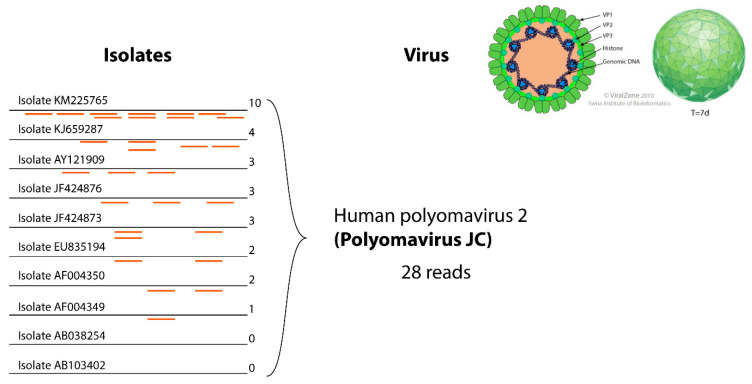
Example of gathering reads for the same virus. In the left part, 10 isolates represent clusters for this virus. Twenty-eight reads show homology to those reference sequences, they can be all grouped under the “human polyomavirus 2” entity, thereby facilitating interpretation of results.

**Figure 4 viruses-12-01248-f004:**
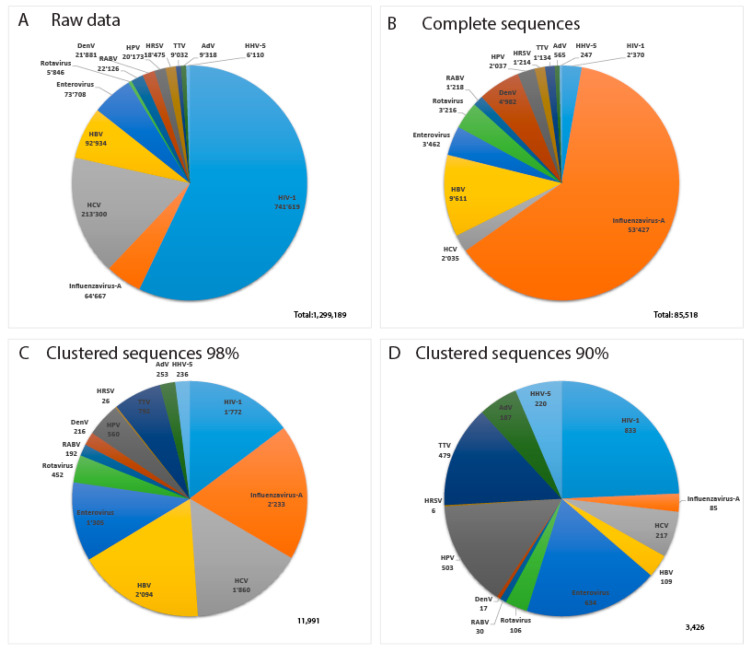
Relative number of sequences for the 13 most sequenced human viruses: (**A**) total sequences from GenBank, (**B**) complete virus sequences, (**C**) Virosaurus 98 and (**D**) Virosaurus 90. (Data from release 2019_10).

**Figure 5 viruses-12-01248-f005:**
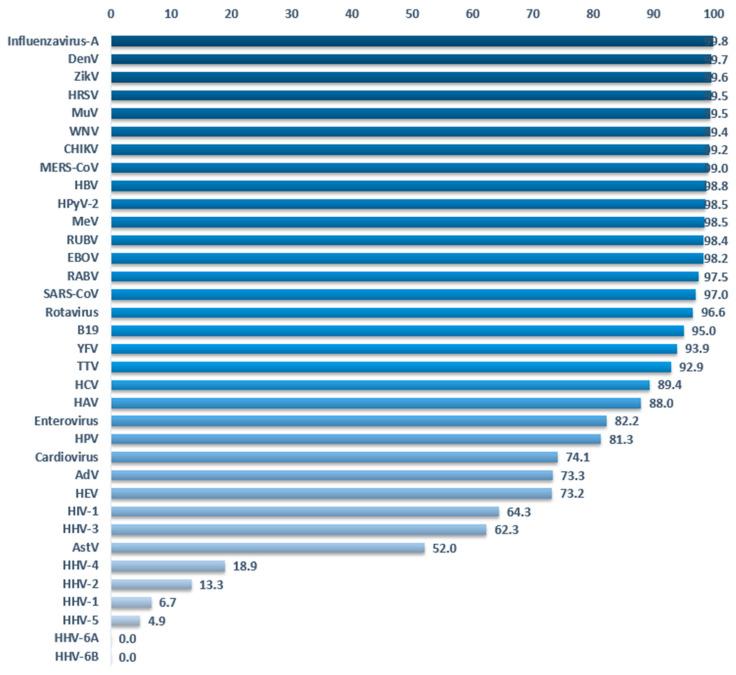
Percentage of sequence reduction by clustering complete genomes at 90%. (Data Virosaurus 2019_10).

**Figure 6 viruses-12-01248-f006:**
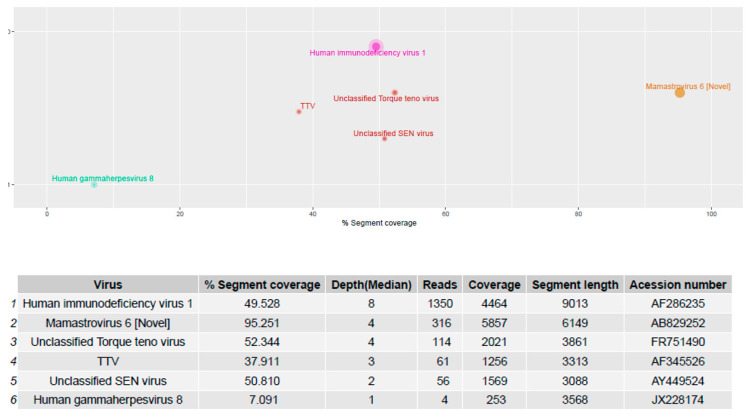
Human blood samples were sequenced and reads generated using the RNA protocol [9] were aligned to the Virosaurus database. The result is easy to interpret and confirms that the patient was positive for a novel human astrovirus, HIV-1 and HHV-8 sequences, as previously reported [23]. Top panel: 2D representation of detected sequences with %segment coverage in the *X*-axis, and depth (median) in *Y*-axis; bottom panel: raw data. Size of dots is relative to number of reads. Anellovirus (TTV) sequences were also detected. The Virosaurus hierarchy allows allocating reads to viral entities: at the level of virus (HIV-1, HHV-8, MastV-6) or higher (TTV). Unclassified SEN viruses are TTV-like genomes. Mamastrovirus (Novel) is a subtyping, allowing differentiating between novel (i.e., MLB and VA/HMO) and classical human astroviruses.

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
