# Peer review of "Virosaurus A Reference to Explore and Capture Virus Genetic Diversity"

_viruses, 2020, doi:10.3390/v12111248_

Round 1
Reviewer 1 Report
I have no further comments on this paper and I am happy for it to be accepted.
Author Response
Thank you for having reviewed the paper.
Reviewer 2 Report
This manuscript reports the development and applications of Virosaurus, a potentially valuable tool for detecting genetic diversity of human viruses. The database was set up to capture virus diversity at the species and genus level. Definition of species and genus differs between plants and animals on one hand and viruses on the other. It is therefore less than ideal to differentiate 'virus species in a looser sense' and 'ICTV species' in this paper, but at least the authors have given a definition of their interpretation. The authors have made an effort to accommodate this reviewer's firm views and that is appreciated. I therefore support publication of the revised paper, so that the Virosaurus is hopefully put to good use for rapid virus diagnosis.
A few minor points: L47 "Lassa virus" should not be in italics; L57: viruses are extremely diverse; L100: "sequences under viral taxonomy" makes no sense.
Author Response
Dear reviewer,
The following changes have been donein the last version:
line 47_ Lassa virus is not in italic anymore,
line 57_ "virus species are extremely diverse, changed to "viruses are extremely diverse"
Line 100: "Some sequences under viral taxonomy" has been changed to ". Some sequences classified under viral taxonomy"
Thank you for reviewing our manuscript.
This manuscript is a resubmission of an earlier submission. The following is a list of the peer review reports and author responses from that submission.
Round 1
Reviewer 1 Report
General comment
In their manuscript entitled “Virosaurus a reference to explore and capture virus genetic diversity”, the authors describe the design of a new virus database named Virosaurus (for Virus Thesaurus), dedicated to the clinical metagenomics studies and diagnosis based on high-throughput sequencing. The main features of this virus database rely on: 1/ the extraction of complete genome sequences of viruses (excepted for herpes and poxviruses) from GenBank (focusing on eukaryotic viruses including those infecting vertebrates, plants and other animals and eukaryotes, 2/ the curation and the annotation of these sequences (in the purpose of clinical use) and 3/ their clustering to reduce the size of the database (to improve its effectiveness for computational analysis).
As underlined by the authors, accurate and up-to-date databases of virus sequences are extremely important for viral metagenomic studies based on HTS data, in order to obtain the most exhaustive and reliable results possible. In this context, the development of a validated database such as Virosaurus can be an interest in the field, especially if this database is automatized and regularly updated.
So I was very interested in this type of development and the subject of this article. But I am ultimately disappointed, both by the quality of the presentation of this article (a lot of repetitions, structure to be reviewed) and its content, which remains rather superficial and therefore the validation part sometimes even includes results already published (see ref. 26).
To make this article more readable and attractive, I suggest a deep rewriting/re-organization of the manuscript and to put more data and emphasis on the "clinical" validation of this database, with the analysis of new/original data in addition of those already published.
I have also different comments regarding the Material and Method section (especially on the methodology for the design of this Virosaurus database) and the Results section which are detailed below:
Major comments
Major comments on Materials and Methods section
One of my main concerns is focused on the selection of the complete genome sequences in the Virosaurus database. Indeed, as indicated lines 100-102, the authors selected the keywords such as “partial”, incomplete”, “near complete”, etc. for this step.
If the authors discarded all the records matching with one of these keywords, they probably excluded many virus species or divergent isolate from a same virus species for which complete sequences of all or part of cds were available. As an example, within the Rhabdoviridae family, many sequences of unique species are not indicated as “complete” in GenBank because the extremities (leader and/or trailer) were not sequenced (which are not very informative for virus detection and discovery) but the rest of the genome is sequenced (95-99%) or because just the cds/gene sequences are available (which are the most interesting). In this aim, more details are needed, especially regarding the sentence lines 102-103.
Another major concern is about updates of the Virosaurus database. Indeed, even if the authors indicate that “The pipeline used to construct Virosaurus is automatized to facilitate regular updates”, they also further indicate that they plan to update the database every year with new sequences (lines 313-314), which probably will considerably limit the interest of potential users. Indeed, discoveries and / or molecular characterization of viruses are increasing almost exponentially. It is therefore probable that between 2 updates (ie 1 year), many sequences of new viruses or viruses not yet characterized will be available, but not included in Virosaurus during this period.
The Materials and Methods section could be improved and/or made clearer. For example:
- Lines 96-97: This step is imprecise. How were refined the quality controls: how many rounds, which kind of clinical samples, etc.)?
- Lines 104-105: What is a “acceptable range”?
- Lines 109-111: Same here: what was this estimation?
- Lines 121-122: What are precisely the list of the controls used, apart size homogeneity (what was the cut-off) and mono-species clustering?
- Lines 128-129: Genotype or serotype data were done only for these 3 viruses? Various other viruses have also such classification which are clinically relevant (such as HIV, adenoviruses, etc.).
- Lines 118-120: The authors describe the use of complete genome for Herpesviridae and Poxviridae but earlier lines 113-115 they describe the use of virus genes. I suggest to reorganize this part.
- Lines 164-165: Why the authors refer to ViralZone and not directly to ICTV, which is the reference of the acronym list?
Major comments on Results section
- The Figure 4C and 4D appear to have been reversed.
- The resolution of all the figures is poor, which make them difficult to read, especially for Figure 3, 4 and 6.
- Figure 6: Additional information is requested in the legend: what does the y-axis correspond to? What does the size of the dots correspond to?
- Line 182: This section contains numerous redundancies with the Materials and Methods section, such as lines 191-194. Some parts will better fit in this section, such as lines 196-198, whereas other parts will better fit in the Discussion section, such as lines 186-189. I suggest to deeply modify the 3.1 section.
- Line 183: The authors indicate that Virosaurus was created using a GenBank release 19-03-2020. However, results presented in Figure 4 and Figure 5 indicated release 2019_10. In addition, the authors present data from the 2020_04 release and also from database 2018_11 (Figure 6). It is confusing and deeply lacks homogeneity.
- Line 200: What does “(Complete = 837909)” refer to?
- Lines 203-204: I do not understand the difference between “complete segments” instead of “complete genomes”. Does it mean that the authors can select for Virosaurus can select part of the segments within the genome of segmented viruses? This was not clearly indicated in the Materials and Methods section. It also means that the coverage of the genome for such viruses will be highly variable, inducing variability in their representativeness, and potential bias.
- Line 297: This section, which should be, from my side, the most interesting and attractive part of this database (i.e. its application to identify viral hits), appears weak, with a validation process done only one a limited number of samples. However, a large quantity of data (NGS type are currently available, which would make it possible to demonstrate more firmly the interest of this database, in comparison with others for example.
Other major comments
I’m not sure that the title of this manuscript “Virosaurus a reference to explore and capture virus genetic diversity” really reflects the application of this database, and can be confusing. Indeed, I do not really agree with the fact that it will “capture virus genetic diversity”, which depend on the level of observation: intrinsic genetic diversity (quasi-species), genetic diversity at the level of a same virus species, etc.
This is especially true with the clustering steps done by the authors. For example, it is probably not possible to exhaustively determine the origin and the diversity of a RABV sequence within the species Rabies lyssavirus based only on 38 complete genome sequences (clustered sequences 98%, figure 4C). It should be true for several other viruses.
I suggest that the authors modify the title accordingly.
Reference citations need to be double check. For example:
- Line 70 (and all over the manuscript): The authors should careful double check the citation order of the references. For example, refs 7-10 are indicated line 70 but the next ref is ref 28 line 103. Then the next reference is ref 20 line 117.
- Line 178: double citation of ref 24.
Minor comments
- Line 39 : I suggest to delete the keyword “NGS” which was not used in the manuscript (or to include in the main text).
- Line 45: I will suggest to use “molecular techniques” instead or before “real-time polymerase chain reaction (PCR)”, the latter being only one of these common techniques, such as conventional PCR. Abbreviation of real-time polymerase chain reaction is qPCR.
- Line 65: Develop/explain “CBER, FDA”.
- Line 92: Add a reference or the related hypertext Internet link for ICTV.
- Line 93: Develop “GAP”.
- Lines 118-119: If the authors refer to the family Herpesviridae and Poxviridae, it should be indicated in italic with a capital letter.
- Line 219: No capital for “Usual”.
- Line 242: No capital for “Genus”.
- Lines 257-262: I suggest to the authors to start the description with 90 than 98 (according to an increasing stringency).
- Line 279: No capital letter for “Dengue” or “Influenza”.
Author Response
We wish to thank the reviewers for the time and efforts spent on our manuscript to help improving it.
Reviewer 1:
Major comments on Materials and Methods section
One of my main concerns is focused on the selection of the complete genome sequences in the Virosaurus database. Indeed, as indicated lines 100-102, the authors selected the keywords such as “partial”, incomplete”, “near complete”, etc. for this step.
If the authors discarded all the records matching with one of these keywords, they probably excluded many virus species or divergent isolate from a same virus species for which complete sequences of all or part of cds were available. As an example, within the Rhabdoviridae family, many sequences of unique species are not indicated as “complete” in GenBank because the extremities (leader and/or trailer) were not sequenced (which are not very informative for virus detection and discovery) but the rest of the genome is sequenced (95-99%) or because just the cds/gene sequences are available (which are the most interesting). In this aim, more details are needed, especially regarding the sentence lines 102-103.
->We consider a genome complete when all CDS are sequenced, regardless of leader and/or trailer completeness for mononegavirales. In our experience, the keywords used to filter out “incomplete” sequences correspond to sequences which do not have all CDS. Nonetheless, I agree that some sequences we did not identify may be missing with these keywords. We will remove these filters in the next release to avoid the risk of missing some sequences. Sentence added line 102: “Complete” is defined here as a sequence containing at least all coding regions.
Another major concern is about updates of the Virosaurus database. Indeed, even if the authors indicate that “The pipeline used to construct Virosaurus is automatized to facilitate regular updates”, they also further indicate that they plan to update the database every year with new sequences (lines 313-314), which probably will considerably limit the interest of potential users. Indeed, discoveries and / or molecular characterization of viruses are increasing almost exponentially. It is therefore probable that between 2 updates (ie 1 year), many sequences of new viruses or viruses not yet characterized will be available, but not included in Virosaurus during this period.
-> We agree with the reviewer, more updates would be better than one per year. The issue here is funding those updates. It is uneasy to find long-term financial support for a database. We will try to update more frequently, but we can only reasonably promise once per year. Nonetheless, in case of a new pandemy or discoveryof important virus we can add patch the database with additional sequence, like we did in January for wuhan ncov.
- Lines 96-97: This step is imprecise. How were refined the quality controls: how many rounds, which kind of clinical samples, etc.)?
->Sentence changed to: All quality controls were refined after testing against at least 10 known blood clinical samples at each release to identify false positives.
- Lines 104-105: What is a “acceptable range”?
->Changed to “range”.
- Lines 109-111: Same here: what was this estimation?
->Added: “Sizes data was establish manually for each virus genus using INSDC data.”
- Lines 121-122: What are precisely the list of the controls used, apart size homogeneity (what was the cut-off) and mono-species clustering?
->Sentence changed to “Clusters were then submitted to controls for size homogeneity (cut-off 80%) or if clusters are mono-species.”
- Lines 128-129: Genotype or serotype data were done only for these 3 viruses? Various other viruses have also such classification which are clinically relevant (such as HIV, adenoviruses, etc.).
->The list of viruses for which we genotyped/serotyped correspond to those of interest for Geneva Hospital clinicians, regarding what they expect to get form a diagnosis.
- Lines 118-120: The authors describe the use of complete genome for Herpesviridae and Poxviridae but earlier lines 113-115 they describe the use of virus genes. I suggest to reorganize this part.
->Part 2.3 and 2.4 were swapped for better coherency.
- Lines 164-165: Why the authors refer to ViralZone and not directly to ICTV, which is the reference of the acronym list?
->Surprisingly ICTV does not provide official acronym names. To our knowledge ICTV website does not offer any structured acronym data. Many of new namings in ICTV do not have yet an acronym. Hence we refer to ViralZone data that was mined from litterature, ICTV approved proposals and various books. This dataset is accessible and maintained.
- The Figure 4C and 4D appear to have been reversed.
->We have switched 4c and 4d in the revised figure.
- The resolution of all the figures is poor, which make them difficult to read, especially for Figure 3, 4 and 6.
->A better figure resolution has been provided for figure 3 and 4. Unfortunately we don’t have better for figure 6.
- Figure 6: Additional information is requested in the legend: what does the y-axis correspond to? What does the size of the dots correspond to?
->Figure 6 and legend have been updated.
- Line 182: This section contains numerous redundancies with the Materials and Methods section, such as lines 191-194. Some parts will better fit in this section, such as lines 196-198, whereas other parts will better fit in the Discussion section, such as lines 186-189. I suggest to deeply modify the 3.1 section.
->The section 3.1 has been revised. The main part of the paper is about the database and how it was created. Indeed there are some redundencies, but we believe it’s important to describe some process in the result section.
- Line 183: The authors indicate that Virosaurus was created using a GenBank release 19-03-2020. However, results presented in Figure 4 and Figure 5 indicated release 2019_10. In addition, the authors present data from the 2020_04 release and also from database 2018_11 (Figure 6). It is confusing and deeply lacks homogeneity.
->There have been several release of Virosaurus datanase, therefore we indicate the release number. All data is available on the website. It may looks to be lacking homogeneity, but actually the database do not change much at each release and all results presented are similar irrespective of the release.
- Line 200: What does “(Complete = 837909)” refer to?
->It’s a mistake, and is now removed.
- Lines 203-204: I do not understand the difference between “complete segments” instead of “complete genomes”. Does it mean that the authors can select for Virosaurus can select part of the segments within the genome of segmented viruses? This was not clearly indicated in the Materials and Methods section. It also means that the coverage of the genome for such viruses will be highly variable, inducing variability in their representativeness, and potential bias.
->For segmented viruses, all known segments are processed by Virosaurus. There are potential bias that only depends on sequence available. Added: “All segments are processed exatly as monopartites genomes for quality and completeness.” In material and methods.
- Line 297: This section, which should be, from my side, the most interesting and attractive part of this database (i.e. its application to identify viral hits), appears weak, with a validation process done only one a limited number of samples. However, a large quantity of data (NGS type are currently available, which would make it possible to demonstrate more firmly the interest of this database, in comparison with others for example.
W ->e agree with the reviewer, but the main authors of this publication have designed and created on the database, and that is the purpose of the manuscript. Virosaurus was mostly validated in a study that will be soon submitted under the title: “Blood Virosphere in Febrile Tanzanian Children”.
I’m not sure that the title of this manuscript “Virosaurus a reference to explore and capture virus genetic diversity” really reflects the application of this database, and can be confusing. Indeed, I do not really agree with the fact that it will “capture virus genetic diversity”, which depend on the level of observation: intrinsic genetic diversity (quasi-species), genetic diversity at the level of a same virus species, etc.
->Virosaurus do not capture quasispecies diversity, but it allows studying the genetic landscape of a virus at the level of a species. The numbers of clusters for a virus gives an insight of its known genetic diversity in INSDC database. Moreover, the whole point of the database is to capture this genetic diversity in a reference database to complement RefSeq, which does not. For example, there are 99 clusters of Lassa virus in Virosaurus90, bur only one sequence in RefSeq. This is why we prefer keeping the title.
Reference citations need to be double check. For example:
- Line 70 (and all over the manuscript): The authors should careful double check the citation order of the references. For example, refs 7-10 are indicated line 70 but the next ref is ref 28 line 103. Then the next reference is ref 20 line 117.
- Line 178: double citation of ref 24.
-> Citations were reprocessed and corrected
- Line 39 : I suggest to delete the keyword “NGS” which was not used in the manuscript (or to include in the main text).
->Done
- Line 45: I will suggest to use “molecular techniques” instead or before “real-time polymerase chain reaction (PCR)”, the latter being only one of these common techniques, such as conventional PCR. Abbreviation of real-time polymerase chain reaction is qPCR.
->Text changed to: or molecular techniques such as real-time polymerase chain reaction (qPCR)
- Line 65: Develop/explain “CBER, FDA”.
->CBER deleted, FDA developed.
- Line 92: Add a reference or the related hypertext Internet link for ICTV.
->Link added
- Line 93: Develop “GAP”.
->It is just about a gap. The word was put in lowercase.
- Lines 118-119: If the authors refer to the family Herpesviridae and Poxviridae, it should be indicated in italic with a capital letter.
->In accordanve with reviewer 2 we changed this names to be: Herpesvirid and Poxvirid,
- Line 219: No capital for “Usual”.
->Done
- Line 242: No capital for “Genus”.
->Done
- Lines 257-262: I suggest to the authors to start the description with 90 than 98 (according to an increasing stringency).
->Done
- Line 279: No capital letter for “Dengue” or “Influenza”.
->done
Reviewer 2 Report
This straightforward paper describes a new database of gene sequences for eukaryotic viruses - Virosaurus - that is maintained at ViralZone. I strongly support the publication of this database as it will greatly assist those working in the area of virus metagenomics. In addition, the ViralZone website is popular and easy to use.
I only have two relatively minor comments:
1. The authors plan to update their Virosaurus annually. While this may be in line with taxonomic updates, I don’t think it is sufficiently frequent to handle the huge new number of viruses that are being generated so rapidly these days. Hence, I think that at least 2 and perhaps 3 updates are needed each year. The more frequently the database is updated, the more useful it will be.
2. There are some minor errors with the English that need to be corrected. The authors just need to give it one more read through.
Author Response
We wish to thank the reviewers for the time and efforts spent on our manuscript to help improving it.
Reviewer 2:
Open Review
- The authors plan to update their Virosaurus annually. While this may be in line with taxonomic updates, I don’t think it is sufficiently frequent to handle the huge new number of viruses that are being generated so rapidly these days. Hence, I think that at least 2 and perhaps 3 updates are needed each year. The more frequently the database is updated, the more useful it will be.
We agree with the reviewer, more frequent updates would be better than one per year. The issue here is funding those updates. It is uneasy to find long-term financial support for a database. We will try to update more frequently, but we can only reasonably promise once per year. Nonetheless, in case of a new pandemy or discovery of important virus we can patch the database with additional sequence, like we did in January for wuhan ncov.
- There are some minor errors with the English that need to be corrected. The authors just need to give it one more read through.
The text has been corrected to improve English.
Reviewer 3 Report
The development and validation of the Virosaurus as reported in this manuscript appears to be an excellent reference database tool to identify human viruses and their genetic diversity in HTS datasets. However, there are 1) significant issues regarding implementation of taxonomic terms ‘species’ and ‘genus/genera’ that contradict ICTV rules and 2) frequent grammatical inaccuracies and mix of present and past tense throughout that are detailed below.
‘Species’ is a taxonomic classification of a virus. A ‘virus’ has a genome, can infect a host and can be detected, whereas a ‘species’ cannot do any of these because it does not exist. According to ICTV rules, virus names are written in lower case and can be abbreviated, while virus species names are written in italics, first letter capitalized and not abbreviated. For example, the virus ‘dengue virus’ (DENV) is taxonomically classified in the species Dengue virus, genus Flavivirus, family Flaviviridae. This should be implemented in the database and throughout the manuscript.
Figure 2 content should be revised accordingly by replacing the term ‘virus species’ with ‘virus’. The genomic data provide a mix of GenBank accession and RefSeq (NC_..) numbers; preferably only GenBank accession numbers should be used because the reference virus name and number may be changed, but not the GenBank accession. What is the “FASTA header” referred to in line 231?
Detailed points:
L31: …identified for viruses of all genera …
L35, L59 and throughout: would “identity” be a better term than “similarity”? Define at first mention what is meant! Also in Fig.1.
L57: give a reference for “several references are necessary to detect all variations within most viruses”
L65 and elsewhere: spell out abbreviations at first use.
L88: revise to “Genera for viruses able to infect …”
L107: what are “ICTV official documents”?
L111: …size ranges for viruses of each genus. The complete… (delete “pool of”)
L119: change to “herpesvirid and poxvirid complete sequences” (taxonomic families do not have sequences); same in Fig.1
L121, L253: reword for better clarity: “submitted to controls”
L136: meaning of “one result”?
L140-143: delete since redundant with the text below
L179: what is the definition of “Go” and “Mo”?
L190: delete “species”
L196-200, 215-217: use past tense
L200: what is “(Complete = 837909)”?
L201: …genomes was more challenging…
L205: …named using official …
L222: Legend to Fig.3: “…reads for the same virus.” This figure does not appear to be referenced in the text.
L242: how are “subspecies” names defined?
L232-245: viruses and species are mixed up in this paragraph that needs a thorough revision
L293: “wild”?
L303: Fig. 6 top panel is missing Y-axis description; what is shown in top vs bottom panel?
Author Response
We wish to thank the reviewers for the time and efforts spent on our manuscript to help improving it.
Reviewer 3:
Open Review
‘Species’ is a taxonomic classification of a virus. A ‘virus’ has a genome, can infect a host and can be detected, whereas a ‘species’ cannot do any of these because it does not exist. According to ICTV rules, virus names are written in lower case and can be abbreviated, while virus species names are written in italics, first letter capitalized and not abbreviated. For example, the virus ‘dengue virus’ (DENV) is taxonomically classified in the species Dengue virus, genus Flavivirus, family Flaviviridae. This should be implemented in the database and throughout the manuscript.
The manuscript has been corrected accordingly.
Figure 2 content should be revised accordingly by replacing the term ‘virus species’ with ‘virus’. The genomic data provide a mix of GenBank accession and RefSeq (NC_..) numbers; preferably only GenBank accession numbers should be used because the reference virus name and number may be changed, but not the GenBank accession.
The figure has been edited. Next release of Virosaurus will put aside NC_XXX numbers and will only use GenBank AC for better coherency.
What is the “FASTA header” referred to in line 231?
This the header part of FASTA file. The sentence has been replaced by : All metadata are added to FASTA files in the header section of each sequence (Figure 2).
Detailed points:
L31: …identified for viruses of all genera …
Corrected.
L35, L59 and throughout: would “identity” be a better term than “similarity”? Define at first mention what is meant! Also in Fig.1.
Corrected as suggested, including figure 1.
L57: give a reference for “several references are necessary to detect all variations within most viruses”
I couldn’t find a reference for that. I know some studies had to include several references, for example lassavirus detection by minion, but this is not explained in the publication.
->the sentence has been removed.
L65 and elsewhere: spell out abbreviations at first use.
Abbreviations have been corrected in the paper.
L88: revise to “Genera for viruses able to infect …”
Done.
L107: what are “ICTV official documents”?
I was referring to ICTV approved proposals documents. Those documents are a mine of information. Text corrected.
L111: …size ranges for viruses of each genus. The complete… (delete “pool of”)
Done.
L119: change to “herpesvirid and poxvirid complete sequences” (taxonomic families do not have sequences); same in Fig.1
Corrected in text and fig 1.
L121, L253: reword for better clarity: “submitted to controls”
Changed to “clusters were reviewed”
L136: meaning of “one result”?
Changed to “one entity”.
L140-143: delete since redundant with the text below
Removed.
L179: what is the definition of “Go” and “Mo”?
Definition added.
L190: delete “species”
Done.
L196-200, 215-217: use past tense
Done.
L200: what is “(Complete = 837909)”?
(Complete = 837909) removed, it was a mistake..
L201: …genomes was more challenging…
Done.
L205: …named using official …
Done.
L222: Legend to Fig.3: “…reads for the same virus.” This figure does not appear to be referenced in the text.
Done, and figure referred in the text.
L242: how are “subspecies” names defined?
Sentence changed to:”Other viruses require subspecies names because of different pathogenicity within the same species.”
L232-245: viruses and species are mixed up in this paragraph that needs a thorough revision
The paragraph has been revised.
L293: “wild”?
Word changed to “field”.
L303: Fig. 6 top panel is missing Y-axis description; what is shown in top vs bottom panel?
Figure 6 and legend have been updated.
Round 2
Reviewer 3 Report
Most minor issues have been adequately addressed by the authors, BUT the authors apparently do not understand the significant difference between a virus (infectious agent) and a species (taxonomic entity). Species is an artificial taxonomic classification for a virus. The ICTV deals with virus species and genus names, but not with the names of viruses. Names for viruses together with suitable abbreviations, are coined by researchers when the virus is first reported. Contrary to the belief of the authors (response to reviewer #1 lines 164-165), it is not surprising that the ICTV does not provide a list of virus names and their acronyms, because ICTV does not deal with virus and their names, only with the taxonomic classification of viruses into “filing boxes” called species which then get classified into bigger evolutionary linked boxes called genera, etc.
There is still a significant number of manuscript locations where “species” term is used instead of “virus”. A few examples: lines 89, 103, 123, 162, 286, Fig.3, 300, 301, 305, 312, 325, 326, 346, 352, 413, 430, 433. The authors should once again read this reviewer’s first comments paragraph and implement changes throughout the manuscript as requested. The paragraph that needed thorough revision in original lines 232-245, now 306-315 is still in very bad shape in the revised version. “Subspecies” should be named “subtype”.
Annoyingly, the authors have now changed names of viruses throughout by writing them in italics. This is wrong. Species names are italisized, not virus names.
Poxvirid and herpesvirid were used only in the bottom part of Fig. 1, while family names are still used in the top of the same figure.
This reviewer did not find the definition of a “complete” sequence in the revised manuscript (see reviewer #1 first author response). In the relevant literature, this is named a “coding complete” sequence; I recommend that this accepted term be used.
There are also several grammatical errors and unclear expression throughout requiring a complete re-write.
Author Response
Dear reviewer,
thank you for pointing out these important problems in the manuscript and allowing us to correct them. I am very sorry not to have well responded and confess that I missed an important point.
Virosaurus sequences have been classified under virus species, because there is no vocabulary controlled names for “virus” in INSDC. Therefore, we tend to speak of “species”, and not virus because this is how sequences are classified, but this is obviously confusing.
>There is still a significant number of manuscript locations where “species” term is used instead of “virus”. A few examples: lines 89, 103, 123, 162, 286, Fig.3, 300, 301, 305, 312, 325, 326, 346, 352, 413, 430, 433.
All page numbers below refers to the revised document ,2nd version, with "all markup" on.
To avoid any confusion, this sentence has been added at the end of introduction line 81
“A ‘virus’ is a biological entity whereas a ‘species’ is a taxonomic classification of a virus. There is no available controlled vocabulary for ‘virus’ names associated with sequences; therefore, we used ‘species’ to classify virus sequences. Throughout the manuscript, we use 'virus species' in a looser sense, and refer to 'ICTV species' wherever applicable. Some ‘virus’ had to be identified for clinical reasons, and are referred to under the term ‘sub-type’.”
Moreover extensive changes have been made throughout the manuscript:
-Line 43: “At least 130 different virus species infecting Humans have been recorded so far”
Changed to “At least 130 different virus infecting Humans have been recorded so far”
-Line 94: “Genera for viruses able to infect eukaryotes as per ViralZone”
changed to “Viruses sequences belonging to genera able to infect eukaryotes as per ViralZone”. We cannot avoid referring to the taxonomy here because the sequences were gathered through Taxonomic IDs of NCBI.
-line 110:” To assess completeness of sequences, a range of sizes was manually curated for each virus genera using all available data in INSDC [17]”
Changed to ” To assess completeness of sequences, a range of sizes was manually curated for each virus using all available data in INSDC [17]”
-line 128: “Clusters were then controlled for size homogeneity (cut-off 80%) and whether clusters contained to a single species.”
Changed to “Clusters were then controlled for size homogeneity (cut-off 80%) and whether clusters referred only to a single species.” I cannot take out species here, because species TaxID are used to classify these sequences.
Line 129: “The few clusters comprising more than one species were checked manually.”
Changed to “The few clusters comprising sequences from more than one species were checked manually.”
Line 132: “Others are true viral species similar enough to others members of the cluster, and for those all species this was reported in the FASTA header representing the cluster. “
Changed to: “Others are sequences similar enough to others members of the cluster but classified under differrent species, and for those all names were reported in the FASTA header representing the cluster. “
Line 147: “Papillomaviruses and torquetenoviruses species were not considered relevant in a clinical report, so for those the usual name is not at the species level, but at genus level in order to pool all these viruses into one entity. “
Changed to: “Papillomaviruses and torquetenoviruses were not considered relevant in a clinical report, so for those the usual name is “HPV” or “TTV” in order to pool all these viruses into one entity. “
Line 159:” Usual name= Name of clinical level entity; If the scientific name is not commonly used, the common clinical name replaces species official name, for example parvovirus B19 is the usual name of the Primate erythroparvovirus 1 species. If clinical level =genus: genus name or acronym, for example all Alphatorquevirus usual names are TTV.”
Changed to :” Usual name= Name of clinical level entity; If the scientific name is not commonly used, the common clinical name replaces it, for example parvovirus B19 is the usual name of the Primate erythroparvovirus 1 species. “
Line 177: “Acronym= Official acronym of the species, as reported in ViralZone acronym list: https://viralzone.expasy.org/resources/Acronyms.xlsx”
Changed to “Acronym= Abbreviation referring to virus name as reported in ViralZone acronym list: https://viralzone.expasy.org/resources/Acronyms.xlsx “
Figure 1: Poxviridae and herpesviridae have been replaced by poxvirid and herpesvirid above virus genes data box.
Line 230: “All sequence taxonomy has been normalized to the species level, thereby “virus species” allows grouping matching reads under one viral entity (Figure 3).”
Changed to: “All sequence taxonomy has been normalized to the species level, thereby matching reads can be classified under one viral entity (Figure 3).”
Figure 3: “species” has been replaced by “virus” in the picture and legend
Paragraph line 256 to 276 rewritten, see below.
Line 285: “Ideally, each cluster should contain only sequences of a single virus species. Clusters comprising more than one species were rare and manually checked.”
Changed to “Ideally, each cluster should contain only sequences belonging to a single species. Clusters comprising sequences from more than one species were rare and manually checked.”
Line 304: “Do we have enough knowledge of each species genetics to detect circulating viruses?”
Changed to “Do we have enough knowledge of each virus genetics to detect circulating viruses?”
Line 312: “A low percentage suggests that we do not have significant data about the genetic landscape of a given species.”
Changed to “A low percentage suggests that we do not have significant data about the genetic landscape of a given virus.”
Line 343: “The Virosaurus hierarchy allows allocating reads to viral entities: species (HIV-1, HHV-8, MastV-6) or genus (TTV).”
Changed to “The Virosaurus hierarchy allows allocating reads to viral entities: at the level of virus (HIV-1, HHV-8, MastV-6) or higher (TTV).”
Line 383: “The dataset of virus genera infecting vertebrate…”
Changed to: “The dataset of virus infecting vertebrate”
The authors should once again read this reviewer’s first comments paragraph and implement changes throughout the manuscript as requested. The paragraph that needed thorough revision in original lines 232-245, now 306-315 is still in very bad shape in the revised version. “Subspecies” should be named “subtype”.
The paragraph have been changed from:
“As virus isolate details are frequently not useful for clinicians in daily routine, the database was designed to allow easy clinical interpretation of viruses at the species level (Figure 3). Thus, we have added annotation about the virus entity: Species name, TaxonomyID, Acronym, usual name, clinical level and typing. The “usual name” displays the common name of a viral species, which can be easier to interpret and more stable than the official species name. For example “Parvovirus B19” has been renamed “Primate erythroparvovirus 1” [18], but most people are not familiar with this latter name. Moreover, some virus genera include many species that are not directly relevant to a clinical diagnostic: there are 29 torque teno virus species named “Torque teno virus 1 to 29” and several of them can infect the same person [19]. For those the common name is “TTV” for all. The same has been done for alphapapillomaviruses, which have all “HPV” as “usual name”. This is indicated in “clinical level”, either “species” or “genus”. Other viruses require subspecies names because of different pathogenicity within the same species. For example, “Enterovirus C” is split into polio and non-polio viruses. The annotated field “clinical typing” comprises genotype (ex: HCV; Norwalk virus), serotype (ex: Dengue virus) or disease (ex: polio, High risk HPV) for some clinically relevant viruses.”
To :(line 256-276)
“Each sequence in INSDC database are linked to a virus isolate. Nonetheless, isolate data are not useful for clinicians in daily routine. To offer relevant results, Virosaurus database was designed to offer a broader classification of viruses at higher level (Figure 3). Ideally sequences would be classified at the ‘virus’ level, but no controlled vocabulary of ‘virus’ names is associated with sequences. Instead, we have used taxonomic classification: all sequences have been classified at the species level and annotation has been added about clinical virus belonging to this taxa: Species name, TaxonomyID, Acronym, usual name, clinical level and typing. The “usual name” displays common virus name, which can be easier to interpret and more stable than the official name. For example parvovirus B19 is the common name of a virus belonging to primate erythroparvovirus 1 [19], most people are more familiar with the first than the latter name. Moreover, some viruses classification is not directly relevant to a clinical diagnostics: there are 29 torque teno virus named Torque teno virus 1 to 29 and several of them can infect the same person [20]. Having a list of TTV was not relevant for clinicians: the entire Torque teno virus received the common name “TTV”. The same has been done for alphapapillomaviruses, which have all “HPV” as “usual name”. These exceptions are signified in the field “clinical level”, describing the taxonomy level for which these names have been assigned to sequences. The annotated field “clinical typing” comprises genotype (ex: HCV; Norwalk virus), serotype (ex: Dengue virus) or disease (ex: polio, High risk HPV) when this data is necessary for some viruses.”
Annoyingly, the authors have now changed names of viruses throughout by writing them in italics. This is wrong. Species names are italisized, not virus names.
Italics have been removed line 143, 145, 255, 256, 258, 262, 263, 264, 301, 302, 319
Poxvirid and herpesvirid were used only in the bottom part of Fig. 1, while family names are still used in the top of the same figure.
Fig 1 has been modified (line 224)
This reviewer did not find the definition of a “complete” sequence in the revised manuscript (see reviewer #1 first author response). In the relevant literature, this is named a “coding complete” sequence; I recommend that this accepted term be used.
It is my mistake, I added a sentence but mixed up copy/paste or saves and it was absent from the revised version. We have the same definition of complete as cited by the reviewer. The sentence has been added line 102: “Complete” is defined here as a sequence containing at least all coding regions.”
There are also several grammatical errors and unclear expression throughout requiring a complete re-write.
The text has been corrected by several scientists and by a medical professional writer. I am out of options to improve the text, but the editor told me that this could be done after if the paper is accepted.